# Emergency Endoscopic Hemostasis for Gastrointestinal Bleeding Using a Self-Assembling Peptide: A Case Series

**DOI:** 10.3390/medicina59050931

**Published:** 2023-05-12

**Authors:** Takashi Murakami, Eiji Kamba, Keiichi Haga, Yoichi Akazawa, Hiroya Ueyama, Tomoyoshi Shibuya, Mariko Hojo, Akihito Nagahara

**Affiliations:** Department of Gastroenterology, Juntendo University School of Medicine, 2-1-1 Hongo, Bunkyo-Ku, Tokyo 113-8421, Japan

**Keywords:** endoscopic hemostasis, emergency endoscopy, gastrointestinal bleeding, self-assembling peptide

## Abstract

*Background and Objectives*: A novel synthetic self-assembling peptide, PuraStat, has been introduced as a hemostatic agent. This case series aimed to evaluate the clinical efficacy of PuraStat for gastrointestinal bleeding during emergency endoscopy. *Cases*: Twenty-five patients with gastrointestinal bleeding who had undergone emergency endoscopy with PuraStat between August 2021 and December 2022 were retrospectively examined. Six patients were receiving antithrombotic agents, and ten patients with refractory gastrointestinal bleeding had undergone at least one endoscopic hemostatic procedure. The breakdown of bleeding was gastroduodenal ulcer/erosion in 12 cases, bleeding after gastroduodenal or colorectal endoscopic resection in 4 cases, rectal ulcer in 2 cases, postoperative anastomotic ulcer in 2 cases, and gastric cancer, diffuse antral vascular ectasia, small intestinal ulcer, colonic diverticular bleeding, and radiation proctitis in each case. The method of hemostasis was only PuraStat application in six cases, and hemostasis in combination with high-frequency hemostatic forceps, hemostatic clip, argon plasma coagulation, and hemostatic agents (i.e., thrombin) in the remaining cases. Rebleeding was observed in three cases. Hemostatic efficiency was observed in 23 cases (92%). *Conclusions*: PuraStat has the expected hemostatic effect on gastrointestinal bleeding during emergency endoscopy. The use of PuraStat should be considered in emergency endoscopic hemostasis of gastrointestinal bleeding.

## 1. Introduction

A novel synthetic self-assembling peptide, PuraStat (3-D Matrix Ltd., Tokyo, Japan), has been introduced as a surgical or endoscopic hemostatic agent [1,2]. PuraStat is indicated for the hemostasis of oozing bleeding in the parenchyma of solid organs, vascular anastomoses, and capillaries of the gastrointestinal tract [2,3,4,5]. The peptide molecule in PuraStat consists of a repeating sequence of three types of amino acids: Arginine, Alanine, and Aspartic Acid, and has a β-sheet structure [3]. The peptide self-assembles into an extracellular scaffold matrix when activated by a pH change associated with exposure to the blood. The matrix sticks to and seals the blood vessels, thereby achieving hemostasis as a mechanical barrier. In addition, the activated matrix promotes tissue proliferation and facilitates effective mucosa healing [2]. Therefore, excluding spurting bleeding, general cases of gastrointestinal bleeding are indicated for hemostasis with PuraStat.

Gastrointestinal bleeding is a medical emergency associated with elevated morbidity and mortality and significant costs to the healthcare system. However, there have been few studies on the efficacy of PuraStat for gastrointestinal bleeding during emergency endoscopic hemostasis. The aim of our study was to assess the safety, efficacy, and technical feasibility of PuraStat as a primary hemostat during emergency endoscopy. Herein, we report a case series of endoscopic hemostasis using PuraStat for gastrointestinal bleeding during an emergency endoscopy.

## 2. Case Descriptions

This retrospective observational study was conducted at the Juntendo University Hospital in Tokyo, Japan. This study included all patients who underwent emergency endoscopic hemostasis using PuraStat for gastrointestinal bleeding between August 2021 and December 2022. Cases in which PuraStat was used during scheduled endoscopic procedures were excluded. Five experienced endoscopists performed all procedures. Patient clinical data, including age; sex; symptoms; underlying disease; use of antithrombotic drugs; necessity of blood transfusion; hemoglobin levels; refractoriness/intractability; causes of gastrointestinal bleeding; and endoscopic procedure data, including the location of the bleeding, types of bleeding, presence or absence of visible vessels, endoscopic hemostasis method, the clinical effectiveness of endoscopic hemostasis, and presence or absence of rebleeding were reviewed. In this study, rebleeding was defined as the development of fresh hematemesis or hematochezia, shock (defined as a systolic blood pressure of ≤90 mmHg or a pulse rate of ≥110 beats per minute) with melena after stabilization, or a drop in hemoglobin of more than 2 g/dL within 24 h. Refractory/intractable bleeding was defined as rebleeding requiring emergency endoscopic hemostasis after failing at least one treatment with endoscopic hemostatic therapy. Clinical effectiveness was defined as achieving no rebleeding within one week after the procedure. All patients were followed up for at least one month after the procedure.

The choice of the hemostasis method was left to the endoscopist. Endoscopic hemostasis using PuraStat was attempted in all cases. When a large amount of blood accumulated in the intestinal tract, it was washed thoroughly with water to identify the bleeding points. PuraStat was applied at the bleeding point using a delivery catheter inserted through the endoscope accessory channel. The procedure was completed after confirming that effective hemostasis was achieved after the wound was covered with a transparent jelly substance of PuraStat.

Twenty-five patients were recruited and treated with PuraStat. The clinical characteristics of the patients are summarized in Table 1. The median age was 73.0 years (range, 40–91 years), and there were twenty men and five women. All but one patient had some underlying medical condition, including cancer, heart disease, diabetes, kidney disease, liver disease, or hypertension, and six were taking antithrombotic drugs. The median hemoglobin level immediately prior to the endoscopic procedure was 7.4 (range 5.0–14.5) g/dL, and 17 patients required red blood cell transfusions. Fifteen of the twenty-five patients had initial bleeding, while the remaining ten patients had refractory or intractable bleeding in which at least one endoscopic hemostasis had been performed before this treatment and resulted in failure.

The endoscopic data of the study patients are summarized in Table 2, and the case numbers in Table 2 correspond to those in Table 1. The breakdown of bleeding was gastroduodenal ulcer or erosion, which was the most common type of bleeding; in 12 cases, bleeding after gastroduodenal or colorectal endoscopic resection was observed in 4 cases, acute hemorrhagic rectal ulcer in 2 cases, postoperative anastomotic ulcer in 2 cases, and gastric cancer, diffuse antral vascular ectasia, small intestinal ulcer, colonic diverticular bleeding, and radiation proctitis in each case. Bleeding occurred in various gastrointestinal tracts, including the stomach, duodenum, small intestine, colon, and rectum. The types of bleeding included oozing, bleeding in 22 cases, and spurting bleeding in 3 cases. Five cases had visible vessels. The method of hemostasis was only PuraStat application in six cases, and hemostasis in combination with high-frequency hemostatic forceps, hemostatic clip, argon plasma coagulation, and hemostatic agents (i.e., thrombin) in the remaining nineteen cases. Rebleeding was observed in only three cases, one of which occurred 10 days after endoscopic hemostasis. Hemostasis was clinically effective in 23 cases (92%), including the aforementioned. All three rebleeding cases underwent endoscopic hemostasis using high-frequency hemostatic forceps and PuraStat and achieved complete hemostasis. No procedure-related side effects were observed in the studied cases.

Figure 1 and Figure 2 show endoscopic images of hemostasis using PuraStat in representative cases.

## 3. Discussion

Endoscopic procedures, surgery, and radiological angiography are the main therapeutic options available for managing gastrointestinal bleeding. Owing to the increasing number of successful endoscopic applications and recent advances in technology, endoscopic procedures are the current standard for hemostasis in patients with gastrointestinal bleeding. In this study, we used PuraStat to treat various gastrointestinal bleeding episodes. The high clinical hemostatic effectiveness of PuraStat was confirmed in patients with underlying disease and/or bleeding diathesis who were taking antithrombotic drugs. It is interesting that hemostatic efficacy was found in 23 out of 25 cases (92%), including 10 cases of refractory or intractable gastrointestinal bleeding. Rebleeding was observed in three cases, while hemostasis was achieved in all cases via endoscopic treatment using PuraStat. de Nucci et al. described the use of PuraStat in 77 patients treated for acute gastrointestinal bleeding, including bleeding occurring as a complication of a previous endoscopic procedure (endoscopic mucosal resection [EMR] and endoscopic retrograde cholangiopancreatography [ERCP]), peptic ulcers, angiodysplasia, cancer, and surgical anastomoses [5]. In their study, the efficacy of PuraStat in primary hemostasis was 90%, and the rebleeding rate was 10%. These findings are consistent with those of this study. In general, the rebleeding rate in patients with peptic ulcers is expected to be 10–20%, and is thought to be two–three times higher in patients taking antithrombotic drugs [6]. In addition, a previous report on acute gastrointestinal bleeding has also shown that rebleeding was significantly associated with patient comorbidities (i.e., cancer, heart disease, diabetes, kidney disease, liver disease, and hypertension), with an odds ratio of 1.2 (95% confidence interval, 1.0–1.4) [7]. The rebleeding rate in this study was lower than estimated, although it could not be compared with that rate. Therefore, endoscopic treatment using PuraStat is expected to have a high hemostatic efficiency. However, further large-scale analyses are required to determine whether PuraStat is effective in patients taking antithrombotic drugs or those with underlying diseases.

Hemostasis with endoscopic clipping or high-frequency hemostatic forceps has been increasingly adopted as a method for hemostasis of the bleeding vessel at the ulcer base. Conventional hemostatic clips achieve hemostasis over the bleeding vessel at the ulcer base through the application of mechanical force between the two jaws. However, secure application of hemoclips to the fibrotic ulcer base can be technically difficult at times [8]. Furthermore, high-frequency hemostatic forceps are endoscopic coagulation devices developed solely for hemostasis. Unlike biopsy forceps, they have a narrow opening angle, small cup, and dull edge to enable pinpoint holding of the target lesion. However, high-frequency coagulation may cause serious complications, such as perforation, due to tissue damage during or after the procedure [9]. In contrast, hemostasis using PuraStat involves only applying PuraStat from a dedicated catheter, which is extremely simple. Subramaniam S et al. used PuraStat in 100 patients undergoing endoscopic resection and stated that only a small amount (mean 1.76 mL) was required for hemostasis, and it took, on average, 69.5 s to stop a bleed [10]. Soons E et al. also reported that it was easy to apply 3 mL of PuraStat to a post-endoscopic mucosal resection defect with a median duration of 2.0 min, and there were no adverse events related to PuraStat application [11]. Similarly, in comparing the conventional hemostatic technique and PuraStat application in patients who developed bleeding associated with endoscopic sphincterotomy, the hemostatic procedure time was significantly shorter in patients using PuraStat, whereas adverse events were significantly less frequently observed in patients using PuraStat [12]. In addition, applying PuraStat does not cause tissue damage due to thermocoagulation; therefore, there is no risk of perforation associated with the procedure. Subramaniam et al. demonstrated that Purastat was an effective hemostat that could reduce heat therapy using coagulation forceps for bleeding during endoscopic submucosal dissection (ESD), resulting in a significant increase in the proportion of patients achieving complete wound healing in the Purastat group compared with controls [13]. Therefore, hemostasis with PuraStat is a simple and safe procedure. The application of PuraStat can be an option for the hemostasis of gastrointestinal bleeding during an emergency endoscopy.

Furthermore, endoscopic hemostasis can be challenging in some scenarios, even for the most experienced endoscopists. The presence of hematic residues and clots in the gastrointestinal tract can pose technical difficulties that may prolong the procedure time or make it impossible [14]. One of the characteristics of PuraStat is that it is a transparent jelly-like substance prone to immiscibility with the blood. Therefore, it is possible to secure a visual field after the application, suggesting that it is easy to identify the bleeding point and confirm hemostasis.

The present study had several limitations. First, it was carried out at a single center and the number of cases was small. Second, PuraStat was not originally indicated for spurting bleeding, although in the present analysis, hemostasis was achieved by applying PuraStat after weakening the bleeding pressure with hemostatic clips or hemostatic forceps, even in three spurting bleeding cases. Third, the method of hemostasis was only PuraStat application in 6 cases, whereas hemostasis in combination with high-frequency hemostatic forceps, hemostatic clips, argon plasma coagulation, and hemostatic agents was used in the remaining 19 cases. It is unclear whether hemostasis was achieved using only the PuraStat or other hemostatic methods. These findings suggest that hemostatic efficiency can be enhanced by combining PuraStat with other hemostatic methods. Fourth, this study was a case series and did not include a control group for comparison. Therefore, further investigation is needed to validate our results and to confirm their clinical effectiveness.

## 4. Conclusions

In this case series, a self-assembling peptide hemostatic hydrogel, PuraStat, was effective in achieving hemostasis of gastrointestinal bleeding during emergency endoscopy. The use of PuraStat should be considered in emergency endoscopic hemostasis of gastrointestinal bleeding.

## Figures and Tables

**Figure 1 medicina-59-00931-f001:**
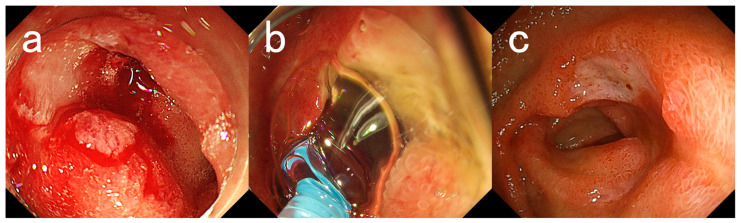
Endoscopic images for Case #13. A 76-year-old woman with a history of myasthenia and diabetes mellitus presented with melena and underwent emergency endoscopy. (**a**) An ulcer with bleeding oozing in the duodenal bulb. (**b**) Application of PuraStat. Hemostasis was achieved by applying 3 mL of PuraStat. (**c**) Endoscopic image taken 2 weeks later. There was a reduction in the size of the ulcer and healing was observed. The patient has not presented with melena for 4 months after her last endoscopy.

**Figure 2 medicina-59-00931-f002:**
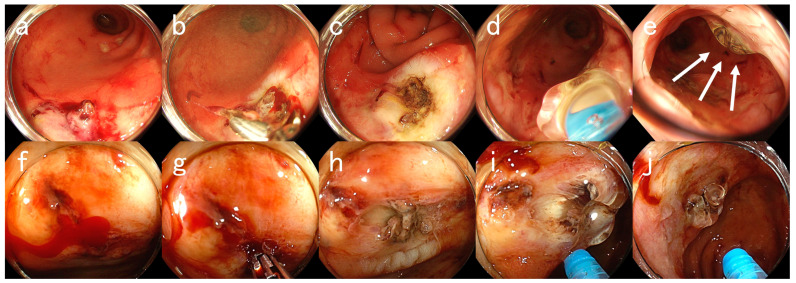
Endoscopic Images for Case #23. An 80-year-old man with a history of diabetes, chronic kidney disease (on hemodialysis), foot gangrene, and hypertension presented with hematochezia. The patient was taking aspirin. (**a**–**e**) First emergency endoscopic views. (**a**) Ulcers with visible blood vessels in the rectum. (**b**) High-frequency hemostasis using hemostatic forceps. (**c**) An ulcer after coagulation. (**d**) Application of PuraStat. (**e**) An ulcer attached with PuraStat (arrows). However, 10 days later, massive hematochezia recurred. (**f**–**j**) Second emergency endoscopy at rebleeding. (**f**) An ulcer with active oozing bleeding in the rectum. (**g**) Re-hemostasis using hemostatic forceps. (**h**) An ulcer after coagulation. (**i**) Application of PuraStat. (**j**) An ulcer attached with PuraStat. Hemostasis was achieved by applying 3 mL of PuraStat. It has been a month since the last endoscopy and the patient has yet to present with hematochezia for one month.

**Table 1 medicina-59-00931-t001:** The Clinical Characteristics of the Studied Patients.

No.	Age	Sex	Symptoms	Underlying Disease	Antithrombotic Drugs	Hemoglobin (g/dL)	Blood Transfusion	Refractoriness /Intractability
1	73	Male	Hematemesis	Chronic heart failure, Arrhythmia, Chronic obstructive pulmonary disease, Hypertension	-	9.4	presence	-
2	82	Female	Hematemesis	Malignant lymphoma	-	7.2	presence	-
3	74	Male	Melena	Parkinson’s disease	-	7.9	presence	-
4	63	Male	Hematemesis	Hypertension	-	9.6	presence	presence
5	61	Male	Hematemesis	Liver cirrhosis	-	5.0	presence	presence
6	77	Male	Melena	Chronic heart failure, Arrhythmia, Hypertension	Edoxaban	7.6	presence	-
7	42	Female	Melena	Liver cirrhosis	-	5.5	presence	-
8	83	Male	Melena	Bile duct cancer, Hypertension	-	6.9	presence	-
9	76	Male	Melena	Prostate cancer, Cerebral infarction, Hypertrophic cardiomyopathy, Hypertension	Warfarin	10.8	-	-
10	69	Male	Anemia	Gastric cancer, Myelodysplastic syndromes, Liver cirrhosis, Portal vein thrombosis	-	6.6	presence	-
11	66	Male	Hematemesis	Chronic kidney disease (on hemodialysis), Hypertension	-	7.4	presence	presence
12	66	Female	Anemia	Liver cancer, Liver cirrhosis, Portal vein thrombosis, Esophageal varices, Hypertension,	-	5.6	presence	presence
13	76	Female	Melena	Myasthenia, Diabetes	-	7.0	-	-
14	77	Female	Melena	Mitochondrial encephalomyopathy, Diabetes, Hypertension	-	6.1	presence	-
15	78	Male	Melena	Colorectal cancer, Arrhythmia, Hypertension	Edoxaban	6.0	presence	presence
16	69	Male	Melena	Dermatomyositis, Interstitial pneumonia, Hypertension	-	10.7	-	-
17	78	Male	Hematochezia	Heart disease, Diabetes, Hypertension	-	6.2	presence	presence
18	50	Male	Hematochezia	Colorectal cancer	-	10.8	-	-
19	40	Male	Melena	-	-	11.2	-	-
20	91	Male	Hematochezia	Cerebral infarction, Myocardial infarction, Arrhythmia, Hypertension	Edoxaban	7.0	presence	presence
21	42	Male	Hematochezia	Ulcerative colitis	-	12.7	-	-
22	53	Male	Hematochezia	Cerebral hemorrhage, Hypertension	Apixaban	8.1	-	-
23	80	Male	Hematochezia	Diabetes, Chronic kidney disease (on hemodialysis), Foot gangrene, Hypertension	Aspirin	6.9	presence	presence
24	78	Male	Hematochezia	Prostate cancer, Hypertension	-	7.4	presence	presence
25	50	Male	Hematochezia	Primary aldosteronism, Hypertension	-	14.5	-	presence

**Table 2 medicina-59-00931-t002:** Endoscopic Data of the Studied Patients.

No.	Causes of Gastrointestinal Bleeding	Location of the Bleeding	Types of Bleeding	Visible Vessels	Method of Hemostasis	Rebleeding	Clinical Effectiveness
1	Gastric cancer	Stomach (cardia)	oozing	-	Hemostatic forceps, PuraStat, Thrombin	-	excellent
2	Gastric ulcer	Stomach (upper body)	oozing	-	Hemostatic forceps, PuraStat	-	excellent
3	Gastric ulcer	Stomach (upper body)	spurting	presence	Hemostatic forceps, PuraStat	-	excellent
4	Bleeding after gastric ESD	Stomach (lower body)	oozing	-	PuraStat	-	excellent
5	Diffuse antral vascular ectasia	Stomach (antrum)	spurting	-	HSE, Hemostatic forceps, Hemostatic clip, PuraStat, Thrombin	-	excellent
6	Gastric ulcer	Stomach (antrum)	oozing	-	PuraStat	-	excellent
7	Gastroduodenal ulcer	Stomach (antrum), Duodenum	oozing	-	Hemostatic forceps, PuraStat, Thrombin	-	excellent
8	Duodenal erosion	Duodenum	oozing	-	Hemostatic clip, PuraStat	-	excellent
9	Duodenal erosion	Duodenum	oozing	-	PuraStat, Thrombin	-	excellent
10	Duodenal erosion	Duodenum	oozing	-	PuraStat, Thrombin	-	excellent
11	Duodenal ulcer	Duodenum	oozing	presence	Hemostatic clip, PuraStat	-	excellent
12	Duodenal ulcer	Duodenum	oozing	-	PuraStat	-	excellent
13	Duodenal ulcer	Duodenum	oozing	-	PuraStat	-	excellent
14	Duodenal ulcer	Duodenum	oozing	-	PuraStat, Thrombin	Presence (2 days later)	poor
15	Duodenal ulcer	Duodenum	oozing	presence	Hemostatic forceps, PuraStat, Thrombin	-	excellent
16	Bleeding after duodenal EMR	Duodenum	oozing	-	PuraStat, Thrombin	-	excellent
17	Postoperative anastomotic ulcer	Ileum	oozing	-	PuraStat	-	excellent
18	Postoperative anastomotic ulcer	Ileum	oozing	-	PuraStat	-	excellent
19	Small intestinal ulcer	Ileum	oozing	-	PuraStat, Thrombin	-	excellent
20	Colonic diverticular bleeding	Ascending colon	oozing	-	Hemostatic clip, PuraStat	-	excellent
21	Bleeding after rectal ESD	Sigmoid colon	oozing	-	PuraStat, Thrombin	-	excellent
22	Acute hemorrhagic rectal ulcer	Rectum	spurting	presence	Hemostatic forceps, PuraStat	Presence (6 days later)	poor
23	Acute hemorrhagic rectal ulcer	Rectum	oozing	presence	Hemostatic forceps, PuraStat	Presence (10 days later)	excellent
24	Radiation proctitis	Rectum	oozing	-	APC, PuraStat	-	excellent
25	Bleeding after rectal EMR	Rectum	oozing	-	PuraStat, Thrombin	-	excellent

APC, argon plasma coagulation; EMR, endoscopic mucosal resection; ESD, endoscopic submucosal dissection; Hemostatic forceps, high-frequency coagulation using hemostatic forceps; HSE, injection of hypertonic saline epinephrine solution; Thrombin, application of liquid thrombin. The case numbers in Table 2 correspond to those in Table 1.

## Data Availability

The datasets of the current study are available from the corresponding author upon reasonable request.

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
