# Peer review of "Emergency Endoscopic Hemostasis for Gastrointestinal Bleeding Using a Self-Assembling Peptide: A Case Series"

_medicina, 2023, doi:10.3390/medicina59050931_

Round 1
Reviewer 1 Report (Previous Reviewer 2)
The authors have revised their manuscript and have responded tocomments correctly. This improved version can be accepted for publication.
Reviewer 2 Report (Previous Reviewer 3)
All issues addressed
Language is acceptable
This manuscript is a resubmission of an earlier submission. The following is a list of the peer review reports and author responses from that submission.
Round 1
Reviewer 1 Report
The authors utilize PuraStat for hemostasis in GI bleeds. However, it is not clear whether hemostasis was achieved due to Purastat or to other agents. This should be clarified and elaborated. PuraStat may be an adjunct to other techniques that may provide hemostasis
Author Response
Thank you for your comments regarding our manuscript. We are truly thankful for the constructive suggestions and have made every attempt to revise our manuscript in accordance with them. The changes that were made follow each comment.
Thank you for your suggestion. We have added the following limitations to the present study:
Third, the method of hemostasis was only PuraStat application in 6 cases, whereas hemostasis in combination with high-frequency hemostatic forceps, hemostatic clips, argon plasma coagulation, and hemostatic agents was used in the remaining 19 cases. It is unclear whether hemostasis was achieved using only the PuraStat or other hemostatic methods. These findings suggest that hemostatic efficiency can be enhanced by combining PuraStat with other hemostatic methods. However, further investigation is needed to validate our results and to confirm their clinical effectiveness.

Reviewer 2 Report
The paper is well written and well organised. It is important to have described the limitations of the study in the final part of the paper. However there are hints for future studies on this haemostatic peptide. The presence of direct oral anticoagulants results in some situations in gastrointestinal haemorrhages that need innovative and valid tools for their resolution in emergencies. The use of PuraStat may be among these tools, although targeted prospective studies with larger populations are needed
Author Response
Thank you for your comments regarding our manuscript. We are truly thankful for the constructive suggestions and have made every attempt to revise our manuscript in accordance with them. The changes that were made follow each comment.
Reply: Thank you for your constructive comment. As per your suggestion, we have added the following sentences to the Discussion section:
Therefore, endoscopic treatment using PuraStat is expected to have a high hemostatic efficiency. However, further large-scale analyses are required to determine whether PuraStat is effective in patients taking antithrombotic drugs or those with underlying diseases.

Reviewer 3 Report
An interesting case series, but not been adequately discussed.
Furthermore the data on this subject is available and more RCTs are required
The following needs to be added to enhance the discussion
a. de Nucci G, Reati R, Arena I, Bezzio C, Devani M, Corte CD, Morganti D, Mandelli ED, Omazzi B, Redaelli D, Saibeni S, Dinelli M, Manes G. Efficacy of a novel self-assembling peptide hemostatic gel as rescue therapy for refractory acute gastrointestinal bleeding. Endoscopy. 2020 Sep;52(9):773-779. doi: 10.1055/a-1145-3412. Epub 2020 Apr 21. PMID: 32316041.
b. Subramaniam S, Kandiah K, Chedgy F, Fogg C, Thayalasekaran S, Alkandari A, Baker-Moffatt M, Dash J, Lyons-Amos M, Longcroft-Wheaton G, Brown J, Bhandari P. A novel self-assembling peptide for hemostasis during endoscopic submucosal dissection: a randomized controlled trial. Endoscopy. 2021 Jan;53(1):27-35. doi: 10.1055/a-1198-0558. Epub 2020 Jul 17. PMID: 32679602.
c. Subramaniam S, Kandiah K, Thayalasekaran S, Longcroft-Wheaton G, Bhandari P. Haemostasis and prevention of bleeding related to ER: The role of a novel self-assembling peptide. United European Gastroenterol J. 2019 Feb;7(1):155-162. doi: 10.1177/2050640618811504. Epub 2018 Nov 5. PMID: 30788128; PMCID: PMC6374844.
d. Soons E, Turan A, van Geenen E, Siersema P. Application of a novel self-assembling peptide to prevent hemorrhage after EMR, a feasibility and safety study. Surg Endosc. 2021;35(7):3564-3571. doi:10.1007/s00464-020-07819-7
e. Uba Y, Ogura T, Ueno S, Okuda A, Nishioka N, Miyano A, Yamamoto Y, Bessho K, Tomita M, Nakamura J, Hakoda A, Nishikawa H. Comparison of Endoscopic Hemostasis for Endoscopic Sphincterotomy Bleeding between a Novel Self-Assembling Peptide and Conventional Technique. Journal of Clinical Medicine. 2023; 12(1):79. https://doi.org/10.3390/jcm12010079
Author Response
Thank you for your comments regarding our manuscript. We are truly thankful for the constructive suggestions and have made every attempt to revise our manuscript in accordance with them. The changes that were made follow each comment.
Response: Thank you for your suggestion. We have added the following sentence to the Discussion section:
de Nucci et al. described the use of PuraStat in 77 patients treated for acute gastrointestinal bleeding, including bleeding occurring as a complication of a previous endoscopic procedure (endoscopic mucosal resection [EMR] and endoscopic retrograde cholangiopancreatography [ERCP]), peptic ulcers, angiodysplasia, cancer, and surgical anastomoses [6]. In their study, the efficacy of PuraStat in primary hemostasis was 90%, and the rebleeding rate was 10%. These findings are consistent with those of this study.
Subramaniam S et al. used PuraStat in 100 patients undergoing endoscopic resection and stated that only a small amount (mean 1.76 mL) was required for hemostasis, and it took, on average, 69.5 sec to stop a bleed [11]. Soons et al. also reported that it was easy to apply 3 mL PuraStat to a post-endoscopic mucosal resection defect with a median duration of 2.0 min, and there were no adverse events related to PuraStat application [12]. Similarly, in comparing the conventional hemostatic technique and PuraStat application in patients who developed bleeding associated with endoscopic sphincterotomy, the hemostatic procedure time was significantly shorter in patients using PuraStat, whereas adverse events were significantly less frequently observed in patients using PuraStat [13]. In addition, applying PuraStat does not cause tissue damage due to thermocoagulation; therefore, there is no risk of perforation associated with the procedure. Subramaniam et al. demonstrated that Purastat was an effective hemostat that could reduce heat therapy using coagulation forceps for bleeding during endoscopic submucosal dissection (ESD), resulting in a significant increase in the proportion of patients achieving complete wound healing in the Purastat group compared with controls [14]. Therefore, hemostasis with PuraStat is a simple and safe procedure. The application of PuraStat can be an option for hemostasis of gastrointestinal bleeding during an emergency endoscopy.
We have responded to the reviewers’ comments in good faith and have changed the original text in accordance with their suggestions. However, we would appreciate any further changes made by the editors, which they may deem necessary. Thank you for your kind patience in evaluating our paper.

Round 2
Reviewer 3 Report
All issues addressed
Author Response
Thank you for taking the time to review despite your busy schedule. We truly appreciate your efforts to provide us with valuable feedback.